# Synthesis, Characterization, and Anticancer Activity of Phosphanegold(i) Complexes of 3-Thiosemicarbano-butan-2-one Oxime

**DOI:** 10.3390/biomedicines11092512

**Published:** 2023-09-12

**Authors:** Sani A. Zarewa, Lama Binobaid, Adam A. A. Sulaiman, Homood M. As Sobeai, Moureq Alotaibi, Ali Alhoshani, Anvarhusein A. Isab

**Affiliations:** 1Department of Chemistry, King Fahd University of Petroleum and Minerals, Dhahran 31261, Saudi Arabia; g202101570@kfupm.edu.sa; 2Department of Pharmacology and Toxicology, College of Pharmacy, King Saud University, Riyadh 11451, Saudi Arabia; 442202864@student.ksu.edu.sa (L.B.); hassobeai@ksu.edu.sa (H.M.A.S.); mralotaibi@ksu.edu.sa (M.A.); ahoshani@ksu.edu.sa (A.A.); 3Core Research Facilities (CRF), King Fahd University of Petroleum and Minerals, Dhahran 31261, Saudi Arabia; 4Interdisciplinary Research Center for Advanced Materials, King Fahd University of Petroleum and Minerals, Dhahran 31261, Saudi Arabia

**Keywords:** ligand, gold(I) complexes, anticancer evaluation, reactive oxygen species, action mechanism

## Abstract

Four novel phosphanegold(I) complexes of the type [Au(PR_3_)(DMT)].PF_6_ (**1**–**4**) were synthesized from 3-Thiosemicarbano-butan-2-one oxime ligand (TBO) and precursors [Au(PR_3_)Cl], (where R = methyl (**1**), ethyl (**2**), tert-butyl (**3**), and phenyl (**4**)). The resulting complexes were characterized by elemental analyses and melting point as well as various spectroscopic techniques, including FTIR and (^1^H, ^13^C, and ^31^P) NMR spectroscopy. The spectroscopic data confirmed the coordination of TBO ligands to phosphanegold(I) moiety. The solution chemistry of complexes **1**–**4** indicated their stability in both dimethyl sulfoxide (DMSO) and a mixture of EtOH:H_2_O (1:1). In vitro cytotoxicity of the complexes was evaluated relative to cisplatin using an MTT assay against three different cancer cell lines: HCT116 (human colon cancer), MDA-MB-231 (human breast cancer), and B16 (murine skin cancer). Complexes **2**, **3**, and **4** exhibited significant cytotoxic effects against all tested cancer cell lines and showed significantly higher activity than cisplatin. To elucidate the mechanism underlying the cytotoxic effects of the phosphanegold(I) TBO complexes, various assays were employed, including mitochondrial membrane potential, ROS production, and gene expression analyses. The data obtained suggest that complex **2** exerts potent anticancer activity against breast cancer cells (MDA-MB-231) through the induction of oxidative stress, mitochondrial dysfunction, and apoptosis. Gene expression analyses showed an increase in the activity of the proapoptotic gene caspase-3 and a reduction in the activity of the antiapoptotic gene BCL-xL, which supported the findings that apoptosis had occurred.

## 1. Introduction

Cancer is currently one of the most significant diseases that threaten the health of people worldwide. Despite the introduction of numerous new cancer diagnoses, the death rate in both industrialized and developing nations is rising [1,2]. Patients who undergo anticancer chemotherapy make up around half of those who are given platinum-based anticancer medications, like cisplatin and its variants [3,4]. Despite some incredible success, platinum-based chemotherapy for cancer is restricted due to its drug resistance and substantial side effects, which include ototoxicity, neurotoxicity, and nephrotoxicity [5,6,7]. These adverse effects result from the innate incapacity of these cytotoxic medicines to distinguish between cancer cells and normal cells. This has motivated researchers to synthesize novel metal-based antitumor agents with various modes of action and coordination characteristics [8].

It has been established that thiosemicarbazones are a significant class of possible binding sites for multidentate compounds for a variety of metal ions [9,10,11]. A five-membered chelate ring or a four-membered ring with high strain is often formed by either the azomethine N or the sulfur group or the hydrazine N atom [11,12,13]. S-monodentate is one of the additional coordination modes seen in thiosemicarbazone complexes (Figure 1) [13]. Thiosemicarbazones are adaptable pharmacophores, due to their ability to form stable compounds with transition metal ions [14,15]. It is well known that thiosemicarbazones have antioxidant, antiviral, and DNA-binding properties [16,17,18,19]. The commercialized methisazone, (trade name Marboran^®^: Burroughs Wellcome & Co., London, England), used to treat smallpox and the anticancer drug 3-aminopyridine-2-carboxaldehyde thiosemicarbazone (trade name Triapine^®^, Vion Pharmaceuticals, Inc., New Haven, CT, USA) have progressed to phase II clinical testing for use against a variety of cancers. The anticancer activity of thiosemicarbazone metal complexes is dependent on the type of tumor cells [19]. It is interesting to note that thiosemicarbazones have been shown to be more active or to have much fewer adverse effects when they are coordinated with metal ions [20]. Thiosemicarbazones and their complexes have cytotoxic activity owing to their capacity to inhibit enzymes, like ribonucleotide reductase, topoisomerase II, and multidrug-resistant protein (MDR1), by interfering with mitochondrial function and releasing reactive oxygen species (ROS) [21,22,23,24,25].

The investigation of gold complexes with various functional ligands that exhibit intriguing physical, chemical, biological, and pharmacological properties has garnered significant interest in recent years [26,27,28,29]. Specifically, gold(I) complexes that contain phosphane ligands have displayed anticancer properties, making them a compelling area of study [29,30,31,32,33]. The literature sources have shown that the incorporation of phosphine ligands into gold(I) complexes can enhance their potency and/or generate selectivity for a range of biological targets. The gold complexes that included phosphine derivatives exhibited better biological responses and higher selectivity indices compared to their nonphosphine counterparts. In this regard, Auranofin has exhibited notable cytotoxic activity in various in vitro and in vivo tumor models, including those that are resistant to cisplatin [34,35,36]. Gold(I)–phosphane complexes have been shown to accumulate selectively in mitochondria, suggesting that this organelle may be their primary site of biological activity. The complexes’ cytotoxic effects appear to be associated with their capacity to bind and inhibit thioredoxin reductase (TrxR), a mitochondrial enzyme, or other thiol-rich proteins and enzymes, leading to their anticancer properties and ability to induce cell apoptosis via reactive oxygen species (ROS) [33,37]. The complexes also effectively inhibit TrxR compared to the free ligands [38].

Tavares and colleagues investigated the cytotoxic properties of gold(I) complexes with aryl-thiosemicarbazones on B16-F10 and CT26.WT tumor cells, as well as BHK-21 nontumor cells. Most of the tested complexes demonstrated moderate cytotoxic activity, with some greater cytotoxicity than cisplatin [38]. González-Barcia and colleagues examined the cytotoxicity of phosphine–thiosemicarbazone gold(I) complexes against several human tumor cell lines, including ovarian adenocarcinoma (MCF-7), cervical epithelial carcinoma (HeLa 229), non-small cell lung cancer (NCI-H460), and normal human lung fibroblast (MRC5) in vitro. The complexes also demonstrated potent inhibition of the activity of thioredoxin reductase and an enzyme crucial for cellular redox signaling [39].

Due to the favorable properties of gold(I) complexes as a viable alternative to Pt agents, coupled with the scarcity of studies regarding gold(I)phosphine complexes with thiosemicarbazone ligands [38,39], and in continuation of our interest in the development of next-generation anticancer gold(I) metallodrugs, herein we report the synthesis and characterization of thiosemicarbazone ligand derived from diacetylmonoxime, and complexation with phosphanegold(I) for the first time in order to develop unique next-generation antitumor metal agents, perhaps maybe with fewer side effects than anticancer medications currently available in the markets. The described complexes **1**–**4** (shown in Figure 1) are characterized by the spectroscopic methods UV-Vis, FTIR, NMR studies, and elemental analyses. In vitro tests on their cytotoxicity against HCT116 (human colon cancer), MDA-MB-231 (human breast cancer), and B16 (murine skin cancer) were evaluated.

## 2. Materials and Methods

### 2.1. General

Diacetlymonoxime and thiosemicarbazide were purchased from Sigma-Aldrich Co., St. Louis, MO, USA. The compounds Bromo(tri-methylphosphine), chloro(tri-ethylphosphine), chloro(tri-tert-butylphosphine), and chloro(tri-phenylphosphine) were purchased from Strems Chemicals, Inc., Newburyport, MA, USA. Dichloromethane and ethanol were provided by Merck KGaA, Darmstadt, Germany and were used directly without further purification. The melting points were measured using a Büchi Melting Point M-560 device from Büchi, Switzerland. Elemental analyses were performed on the PerkinElmer Series II (CHNS/O) Analyzer 2400 series, Shelton, CT, USA. ^1^H and ^13^C NMR spectra were acquired on a Bruker 400 spectrophotometer, MA, USA., while ^31^P NMR spectra were acquired on JEOL 600, JASTEC Superconductor, Hyogo, Japan. On a NICOLET 6700 FTIR, Thermo Electron Corporation, Madison, WI, USA, employing potassium bromide (KBr) pellets over range of 400–400 cm^−1^. The solid-state FTIR spectra of free ligands and their phosphanegold(I) complexes were collected.

### 2.2. Synthesis of Ligand (TBO)

The ligand (**TBO**) was prepared by adding an ethanolic solution (20 mL) of diacetylmonoxime (101.105 mg, 1 mmol) to a well-stirred warm ethanolic solution (20 mL) of thiosemicarbazide (91.14 mg, 1 mmol) with a few drops of glacial acetic acid. The solution was refluxed for four hours at 80 °C. The reaction mixture was then concentrated to half and cooled down to room temperature to afford the crystalline product, which was re-crystallized and collected by filtration and dried in a desiccator over CaCl_2_ (Figure 2).

**Ligand (TBO):** Yield = 0.176 g (92%). M.P: 202.5 °C. Anal. Calcd. for C_5_H_10_N_4_SO: MW g/mol C, 34.47; H, 5.79; N, 32.16; S, 18.40. Found: C, 34.75; H, 5.39; N, 31.50; S, 18.22. IR (KBr, cm^−1^): 3415 v(OH), 3264, 3238 ν(NH_2_), 3156 ν(N–H), 1598 ν(C=N–OH), 1494 ν(C=N), 1374, 1286, 845 ν(-N–C=S). ^1^H NMR (400 MHz, DMSO) δ 11.57 (OH, s, 1H), 10.19 (NH, s, 1H), 8.34 (NH_2_, s, 1H), 7.75 (NH_2_, s, 1H), 2.08 (H1’, s, 3H), 1.99 (H2’, s, 3H). ^13^C NMR (101 MHz, DMSO) δ 179.00 (C=S), 154.72 (C=N–OH), 147.46 (C=N–N), 11.76 (C1’), 9.38 (C2’).

### 2.3. Synthesis of [Au(PR_3_)(TBO)]PF_6_ Complexes (***1***–***4***)

Complexes **1**–**4** were prepared by combining AgPF_6_ (0.126 g, 0.5 mmol) dissolved in 10 mL ethanol with Me_3_PAuBr(**1**) (0.176 g, 0.5 mmol), Et_3_PAuCl(**2**) (0.175 g, 0.5 mmol), t-But_3_PAuCl(**3**) (0.217 g, 0.5 mmol), and Ph_3_PAuCl(**4**) (0.247 g, 0.5 mmol) in 10 mL dichloromethane. The mixture was stirred at room temperature for 15 min and filtered off to remove AgBr and AgCl as yellow and white precipitates, respectively. The filtrate of each complex was then added to respective ligand **TBO** (0.087 g, 0.5 mmol) dissolved in 10 mL ethanol. After two hours of stirring, the solution was filtered. The pale-yellow solutions were kept at room temperature. After some days, white to yellow crystals were obtained. The obtained crystals were recrystallized from a mixture of acetone/acetonitrile (1:1).

**[(Me)_3_PAu(TBO)].PF_6_ 1:** Yield: 0.14 g (47%) M.P: 141–145 °C. Anal. Calcd. for C_8_H_19_AuF_6_N_4_P_2_SO: MW = 592.23 g/mol: C, 16.22; H, 3.23; N, 9.46; S, 5.41. Found: C, 15.79; H, 2.97; N, 9.02; S, 5.14. IR (KBr, cm^−1^): 3427 v(OH), 3248, 3222 ν(NH_2_), 3165 ν(N–H), 1605 ν(C=N–OH), 1507 ν(C=N), 1365, 1294, 845 ν(-N–C=S). ^1^H NMR (400 MHz, DMSO) δ 11.84 (OH, s, 1H), 10.94 (NH, s, 1H), 9.09 (NH_2_, s, 1H), 8.40 (NH_2_, s, 1H), 2.14 (H1’, s, 3H), 2.02 (H2’, s, 3H), 1.63 (H1, d, *J* = 11.9 Hz, 6H). ^13^C NMR (101 MHz, DMSO) δ 174.00 (C=S), 154.59 (C=N–OH), 152.77 (C=N–N), 14.71, 14.33 (C1), 12.38 (C1′), 9.52 (C2’). J-value ^13^C NMR (101 MHz, DMSO) δ 14.52 (C1, d, *J* = 38.5 Hz). ^31^P NMR (600 MHz, DMSO) δ 0.089, −143.591 (PF_6_).

**[(Et)_3_PAu(TBO)].PF_6_ 2:** Yield = 0.19 g (59%). M.P: 157–160 °C. Calc. for C_11_H_25_AuF_6_N_4_P_2_S: MW = 634.31 g/mol: C, 20.83; H, 3.97; N, 8.83; S, 5.06. Found: C, 20.77; H, 3.85; N, 9.06; S, 5.13. IR (KBr, cm^−1^): 3429 v(OH), 3247, 3220 ν(NH_2_), 3165 ν(N–H), 1607 ν(C=N–OH), 1508 ν(C=N), 1367, 1297, 842 ν(-N–C=S). ^1^H NMR (400 MHz, DMSO) δ 11.76 (OH, s, 1H), 10.80 (NH, s, 1H), 8.96 (NH_2_, s, 1H), 8.28 (NH_2_, s, 1H), 2.13 (H1’, s, 4H), 2.02 (H2’, s, 3H), 1.94 (H1, d, *J* = 7.7 Hz, 7H), 1.12 (H2, d, *J* = 19.1 Hz, 10H). ^13^C NMR (101 MHz, DMSO) δ 174.04 (C=S), 154.50 (C=N–OH), 152.45 (C=N–N), 16.97, 16.61 (C1), 12.28 (C1’), 9.42 (C2’), 8.97 (C2). J-value: ^13^C NMR (101 MHz, DMSO) δ 16.79 (C1, d, *J* = 35.3 Hz), 9.20 (d, *J* = 45.4 Hz). ^31^P NMR (600 MHz, DMSO) δ 36.899, −143.591 (PF_6_).

**[(t-But)_3_PAu(TBO)].PF_6_ 3:** Yield = 0.25 g (69%). M.P: 168.5–169.3 °C. Calc. for C_17_H_37_AuF_6_N_4_P_2_S: MW = 718.47 g/mol: C, 30.47; H, 2.94; N, 4.30; S, 4.93. Found: C, 31.46; H, 2.21; N, 3.72; S, 4.55. IR (KBr, cm^−1^): 3429 v(OH), 3248, 3220 ν(NH_2_), 3165 ν(N–H), 1607 ν(C=N–OH), 1508 ν(C=N), 1370, 1297, 838 ν(-N–C=S). ^1^H NMR (400 MHz, DMSO) δ 11.79 (OH, s, 1H), 10.95 (NH, s, 1H), 9.11 (NH_2_, s, 1H), 8.42 (NH_2_, s, 1H), 2.14 (H1’, s, 3H), 2.02 (H2’, s, 3H), 1.48 (H2, d, *J* = 13.9 Hz, 27H). ^13^C NMR (101 MHz, DMSO) δ 174.02 (C=S), 154.50 (C=N–OH), 152.08 (C=N–N), 31.80 (C2), 29.55 (C1), 12.26 (C1’), 9.41 (C2’). ^31^P NMR (600 MHz, DMSO) δ 97.021, −143.591 (PF_6_).

**[(Ph)_3_PAu(TBO)].PF_6_ 4:** Yield = 0.22 g (56%). M.P: 181–183. Calc. for C_17_H_37_AuF_6_N_4_P_2_S: MW = 778.44 g/mol: C, 30.47; H, 2.94; N, 4.30; S, 4.93. Found: C, 31.46; H, 2.21; N, 3.72; S, 4.55. IR (KBr, cm^−1^): 3428 v(OH), 3246, 3221 ν(NH_2_), 3158 ν(N–H), 1605 ν(C=N–OH), 1507 ν(C=N), 1367, 1295, 841 ν(-N–C=S). ^1^H NMR (400 MHz, DMSO) δ 11.67 (OH, s, 1H), 10.55 (NH, s, 1H), 8.72 (NH_2_, s, 1H), 8.06 (NH_2_, s, 1H), 7.74–7.19 (H2, H3, H4, m, 15H), 2.10 (H1’, s, 3H), 2.01 (H2’, s, 3H). ^13^C NMR (101 MHz, DMSO) δ 176.59 (C=S), 154.60 (C=N–OH), 149.73 (C=N–N), 133.99, 133.86, 132.39, 129.77, 129.66 (C1–C4, 12.00, 9.38. ^31^P NMR (600 MHz, DMSO) δ 36.792, −143.591 (PF_6_).

### 2.4. Cell Culture

HCT116, MDA-MB-231, and B16 cells were acquired from American Type Culture Collection (ATCC, Manassas, VA, USA). They were grown in DMEM, 10% fetal bovine serum (FBS), and 1% Penicillin/Streptomycin. All cells were incubated in a humidified chamber at 37 °C with 5% CO_2_.

### 2.5. Cell Viability Assay

To evaluate the cell viability effect of novel complexes (**TBO**, **1**, **2**, **3**, and **4**) on cancer cell lines HCT116 (human colon cancer), MDA-MB-231 (human breast cancer), and B16 (murine skin cancer), 1 × 10^4^ cells were seeded with 200 µL culture media in each well of a 96-well plate and incubated for 24 h. Next, the media were removed, and cells were exposed to different concentrations (1, 3, 10, 30, and 100 µM) of each complex, cisplatin (a positive control), and untreated cells as negative control. Then, the plates containing the drugs were incubated for 24 h. Later, 15 µL serum-free medium containing 5 mg/mL MTT dye (3-(4,5-dimethylthiazol-2-yl)-2,5-diphenyltetrazolium bromide) was added to each well and incubated for 2 h in the CO_2_ incubator at 37 °C. The formazan crystals were dissolved in 200 µL of isopropyl alcohol after removing MTT dye and drug-containing media. The absorbance indicating cell viability was measured using a microplate reader at 570 nm wavelength. The following formula was used to calculate the percentage of cell viability: Cell viability % = 100 × (AbsorbanceCompound)/(AbsorbanceControl). The IC50 value of the novel groups and cisplatin were calculated in μM using Excel and Graph-Pad Prism version 9.3 (Graph-Pad software, San Diego, CA, USA).

### 2.6. Mitochondrial Membrane Potential (ΔΨ_m_) Assay

A total of 1 × 10^4^ cells of MDA-MB-231 were cultured in 200 µL culture medium per well in a 96-well plate. Concentrations of 0.3, 1, 3, and 5 µM of the novel complex **2** and control wells were used in this experiment to determine the shift in mitochondrial depolarization. The 96-well plate was incubated for 24 h. Afterward, with the use of the JC-1 apoptosis detection kit, cells were incubated for 30 min. ΔΨ_m_ alterations were measured by red/green fluorescence emission with excitation/emission 560/595 nm and 485/535 nm, respectively. Visualization of fluorescence was by the Synergy H1 fluorescent microplate reader. Quantification of the ratio of J-aggregate to JC-1 monomer intensity was calculated by counting for loss in ΔΨ_m_ as a decrease in the of aggregate/monomer ratio, while an increase in ΔΨ_m_ implied an increase in the ratio.

### 2.7. ROS Assay

A total of cells 1 × 10^4^ MDA-MB-231 were cultured in 200 µL culture medium per well in 96-well plates. Then, they were treated at concentrations of 0.3, 1, and 3 µM of complex **2**, and control wells were replaced with fresh media. The plates were incubated for 1–2 h, and 50 μL of 25 µM concentration of the fluorescent dye 2′,7′-dichlorodihydrofluorescein diacetate (H_2_DCFDA) was added to each well and incubated for 30 min at 37 °C in 5% CO_2_ humidity. A total of 500 μM of H_2_O_2_ was used as positive control. The plates were read with Synergy H1 fluorescent microplate reader (Thermofisher, Waltham, MA, USA) with excitation of 485 nm/emission 535 nm.

### 2.8. Quantitative Reverse Transcription-Polymerase Chain Reaction (qRT-PCR)

The extraction of total RNA was carried out with RNAbler Kit (Haven Scientific, Thuwal, Saudi Arabia) as indicated by the protocol. RNA concentration and purity, at a 260/280 ratio of ~2.0, were measured by nanodrop (NanoDrop Lite Plus, Thermo Scientific, USA). Later, the extracted RNA was reverse-transcribed into cDNA using the High-Capacity cDNA reverse transcription kit (Applied Biosystems^®^; Thermo Fisher Scientific, Inc. USA) in a thermocycler. The second step of qRT-PCR was by using a QuantStudio 6 Flex system (Thermo Fisher Scientific, Inc., Franklin, MA, USA) for qPCR reactions with the use of EverGreen 2× PCR Master Mix (Haven Scientific, Thuwal, Saudi Arabia). GAPDH was served as the housekeeping gene for the qPCR step. The thermocycling conditions were as follows: 95 °C for 10 s; 60 °C for 20 s; and 72 °C for 10 s and a total of 40 cycles. Calculation of the relative gene expression was conducted using the comparative cycle threshold (CT) (2^−ΔΔCT^) method. The specific forward and reverse primer sequences (synthesized and purchased from Invitrogen) were as follows:

GAPDH: forward 5′-AGC CAC ATC GCT CAG ACA C-3′ and reverse 5′-GCC CAA TAC GAC CAA ATC C-3′.

BCL-xL: forward 5′-CTG AAT CGG AGA TGG AGA CC-3′ and reverse 5′-TGG GAT GTC AGG TCA CTG AA-30′.

Caspase-3: forward 5′-GAG TGC TCG CAG CTC ATA CCT-3′ and reverse 5′-CCT CAC GGC CTG GGA TTT-3′.

### 2.9. Solution Chemistry of Complexes (***1***–***4***)

To investigate the stability of complexes **1–4** recorded on GENESYS 10S UV-Vis spectrophotometer, Thermo Scientific, Franklin, MA, USA. The complexes were dissolved separately in DMSO and a solution containing EtOH:H_2_O (1:1) and subsequently analyzed for stability during period of 0 h, 12 h, 24 h, and 48 h.

## 3. Results

### 3.1. Preparation and FT-IR Characterization

The synthesis of the complexes (**1**–**4**) involved a two-step chemical process. In the initial stage, the starting complexes chloro(tri-methylphosphine)gold(I), chloro(tri-ethylphosphine)gold(I), chloro(tri-tert-butylphosphine)gold(I), and chloro(tri-phenylphosphine)gold(I) were mixed with silver hexafluorophosphate in a 1:1 molar ratio. Then, an equimolar amount of TBO ligand was added after filtering off the AgCl precipitate. The resulting [Au(PR_3_)(TBO)]PF_6_ complexes (**1**–**4**) were isolated as dry crystalline solids. Based on elemental analyses results, the stoichiometry of the complexes was established. The results of the elemental analyses for the free ligand (TBO) and the gold(I) complexes **1**–**4** are listed in Appendix A. Complexes possess stability against air and heat, exhibiting a melting point in the range of 140–180 °C.

The IR characteristic wavenumber peaks for free ligand (TBO) and its complexes (**1**–**4**) are summarized in Appendix A, while the FT-IR spectra of the compounds are provided in Appendix A. The bonding between the ligand and the metal ion is determined by comparing the IR spectra of the free ligand and its complexes. In the IR spectrum of the ligand (TBO), a distinct peak at 3415 cm^−1^ was observed, indicating v(OH) stretching vibrations. The bands of ν(NH_2_) and ν(NH) appeared at 3264/3238 and 3156 cm^−1^, respectively [40]. Additionally, two other bands detected at 1598 and 1494 cm^−1^ were attributed to ν(C=N–OH) and ν(C=N), respectively [41,42]. The results confirm the synthesis of the TBO ligand. Comparing the IR data of the complexes (**1–4**) to that of the free ligand, the frequency of the v(NH_2_) group in the complexes shifted toward lower frequencies by 16, 17, or 18 cm^−1^ due to the interaction between the NH and NH_2_ groups, leading to two distinct forms of intra- and intermolecular hydrogen bonding involving imine groups. Similar structures of complexes have been documented in the literature [40,43]. It was observed that the position of the thioamide ν(HN–C=S) bands of thiosemicarbazone appeared at 1374, 1286, and 845 cm^−1^, which are the result of mixed vibrations that combine C–N stretching, C=S stretching, and NH bending [44,45]. The frequencies shifted 4–10 cm^−1^ toward lower wave numbers in the complexes (as shown in Appendix A). This shift indicates that the sulfur of thione is involved in the coordination with the metal ion.

### 3.2. NMR Characterization

The NMR spectra of the free ligand (TBO) and its complexes were carried out in a DMSO-d_6_ solution. The ^1^H NMR chemical shifts of both the free ligand and its corresponding complexes (**1**–**4**) are given in Table 1. The ^1^H NMR data of free ligand (TBO) revealed signals at 2.08 and 1.99 ppm (methyl protons H1’, H2’); 7.75 and 8.34 ppm (NH_2_); 10.19 ppm (NH); and 11.57 ppm (OH). The two distinct singlet signals for –NH_2_ proton indicate that the partial double-bond nature prevents the C–N bond from revolving freely, which is shown in (Appendix A). In the ^1^H NMR chemical shifts of gold(I) complexes (**1**–**4**), the signal of the OH group appeared at 11.67–11.84 ppm appeared relatively downfield with a little shift in the range of 0.27–0.1 ppm with respect to free ligand (TBO), indicating the protonated nature of the C=N–OH group. A little downfield shift in the (NH and NH_2_) peaks of the complexes at 10.55–10.94 ppm for (NH) and at 8.72–9.1/8.06–8.42 ppm for (NH_2_) vs. free ligand that observed at 10.19 ppm for (NH) and 7.75/8.34 ppm for (NH_2_). The observed downfield shifts indicating the coordination of the gold(I) center to the sulfur of the thione group. The little shifts in protons of the NH_2_ group is attributed to a potential hydrogen bonding between the amino group and solvent (NH_2_…DMSO) [38,45,46].

^13^C NMR chemical shift data are presented in Table 2. The ^13^C NMR data of ligand TBO showed peaks at 9.38 (C1’), 11.76 (C2’), 147.46 (C=N–N), 154.72 (C=N–OH), and 179.00 ppm (C=S), which are in accordance with the literature [45,47,48]. In the gold(I) complexes (**1**–**4**), the C=N–N signals appeared in the range of 149.73–152.77 ppm compared to the free ligand at 147.46 ppm. The up-shift of C=S in the complexes proved the coordination of the thione sulfur to the gold center, which was identified at 179 ppm in the free ligand (TBO). The ^31^P NMR data of complexes (**1–4**) showed single resonance at 0.089 ppm (**1**), 36.98 ppm (**2**), 97.0 ppm (**3**), and 36.79 ppm (**4**) with respect to the precursors appeared at −9.66 ppm (**1**), 32.0 ppm (**2**), 91.15 ppm (**3**), and 33.8 ppm (**4**). The downfield shift in the range of (2.974–9.751) ppm compared to the phosphane of the precursors is shown in Appendix A. ^31^P NMR data confirmed the synthesized complexes and are in agreement with (^1^H, ^13^C) NMR and IR data. ^1^H, ^13^C, and ^31^P NMR spectra are given in Appendix A.

### 3.3. Evaluation of Novel Complexes (***1***–***4***)

The cell viability assessment of novel complexes (TBO, **1**, **2**, **3**, and **4**) was accomplished by MTT assay. Three cell lines: HCT-116, MDA-231, and B16 were exposed to the five novel complexes, in addition to cisplatin as a positive control, and were monitored for changes in their cellular metabolic activity. The intensification of the MTT signal is interpreted as an increase in metabolic activity and vice versa.

Table 3 represents the relative IC_50_ values of the novel complexes (**1**–**4**). The lower IC_50_ values compared to cisplatin establish the potential cytotoxic complexes as effective anti-cancers, which are the three prospective complexes **2**, **3**, and **4**. All three cell lines (HCT116, MDA-MB-231, and B16) are represented with their percentages of cell viability in Figure 2. HCT116 cells showed enhanced cytotoxic activity of **2**, **3**, and **4** in higher concentrations (30 and 100 µM) compared to cisplatin (Figure 2a). In addition, the same complexes **2**, **3**, and **4** were exhibiting higher cytotoxic activity compared to cisplatin against MDA-MB-231 and B16, which were evident in the 10 µM concentrations and higher (Figure 2b,c). Complex **2** showed a decrease in cell viability starting from 3 µM, with a substantial decline with increased concentrations. This can be an indication of complex **2** as a prospective anticancer agent. Further in vivo studies should be considered for the evaluation of the side-effects profile on major organs and the efficacy properties as anticancer therapy.

### 3.4. Mitochondrial Damage and Induction of Apoptosis by the Novel Complex 2

To study the cytotoxic effects of complex **2** on breast cancer cells by defining apoptosis using JC-1 staining assay, ΔΨm disturbance was measured to indicate that MDA-MB-231 cells showed a drop in the aggregate/monomer ratio with increased complex **2** concentrations. The results indicate that MDA-MB-231 cells treated with the novel complex **2** demonstrated a continuous loss of ΔΨm with increased concentrations and a significant decrease in the red/green fluorescence intensity ratio in the 3 and 5 µM values (*p* < 0.05) compared with the control (Figure 3). The decline in ΔΨm can be interpreted as a dose-dependent loss due to mitochondrial depolarization induced by complex **2**. Therefore, the activation of the caspase cascade and the occurrence of apoptosis can be a subsequent event after ΔΨm disturbance.

### 3.5. Induction of Mitochondrial ROS and Apoptosis by Complex ***2***

Elevation in intracellular ROS levels leads to apoptosis by inducing oxidative stress [49]. To accomplish the relationship between the alterations in ROS and complex **2**-induced apoptosis, H2DCFDA dye was used. The assay significantly increased fluorescence intensity in the 0.3, 1, and 3 µM concentrations of the novel complex **2** and the positive control H_2_O_2_ (*p* < 0.05). The fluorescence intensity indicates higher ROS levels. These observations support the notion that the novel complex **2** can effectively induce apoptosis by ROS in breast cancer cells. (Figure 4).

### 3.6. Complex 2 Induction of Apoptosis by Enhancing the Production of Proapoptotic Gene Expression and Reducing Antiapoptotic Genes in MDA-MB-231 Cells

To understand the molecular mechanism in which the novel complex **2** works, we examined its effects on apoptosis. The quantification of apoptotic genes, with respect to GAPDH as the control, revealed changes in the expression of the proapoptotic gene caspase-3 and antiapoptotic gene BCL-xL. The MDA-MB-231 cells were treated with 0.3, 1, and 3 µM concentrations of complex **2** for 24 h; next, the extracted RNA of the treated cells was analyzed. The caspase-3 gene expression levels increased significantly under the complex **2** treatment (*p* < 0.05). As seen in Figure 5a, caspase-3 gene expression levels showed a steady increase, indicating the apoptotic death occurring with increased complex **2** concentrations.

In order to determine the involvement of the antiapoptotic gene BCL-xL, qPCR measurements were taken of BCL-xL in the MDA-MB-231 cell line exposed to different concentrations of complex **2** for 24 h incubation (Figure 5b). There was a slight increase in the level of BCL-xL in the 0.3 µM concentration. In addition, both 0.3 and 1 µM concentrations did not show significant increase after 24 h complex **2** exposure (*p* > 0.05). Whereas, in the 3 µM concentration, complex **2** showed a significantly higher expression compared to the control (*p* < 0.05). The upregulation of the proapoptotic gene caspase-3 as well as the downregulation of the antiapoptotic gene BCL-xL contributes to cell destruction by programmed cell death and mitochondrial dysfunction [50], which was observed after complex **2** treatment.

### 3.7. Solution Chemistry of Complexes (***1***–***4***)

The stability of complexes **1**–**4** was assessed by subjecting them to UV-visible spectroscopy at room temperature in DMSO and EtOH:H_2_O (1:1) solutions. The complexes exhibited complete solubility in the respective solutions. Spectral measurements were taken at various time intervals ranging from 0 to 48 h, as illustrated in the Appendix A. Throughout the experiments, no changes were observed in the spectra at different time intervals. There were no noticeable shifts, either red or blue, in the absorption peaks of each complex, nor were there any new absorption peaks observed. These findings strongly indicate that complexes **1**–**4** remained stable without undergoing decomposition throughout the 48 h duration of the study.

## 4. Discussion

We investigated the novel complexes (TBO, **1**, **2**, **3**, and **4**) with cell viability assay in three cell lines: HCT116, MDA-MB-231, and B16 and compared them with cisplatin. The IC_50_ values were obtained for each complex. Complexes **2**, **3**, and **4** exhibited significant effects on each tested cancer cell. The cell proliferation inhibition exhibited by the novel complexes is represented in a dose-dependent manner shown in Figure 2. Similar observations were documented in our previous study that showed the effect of novel gold(III) complexes compared to cisplatin against multiple cell lines (A549, HeLa, MDA-231, and MCF-7), whereas those complexes caused a reduction in cell viability with increased complex concentrations [51].

Due to the novelty of the complexes, their underlying molecular mechanisms are unknown. Therefore, we report here that complex **2** exhibits strong anticancer activity in breast cancer cells via the induction of oxidative stress, mitochondrial dysfunction, and apoptosis. Apoptosis is a form of programmed cell death, which is a process that happens in normal tissue and as a stress response to signals, such as reactive oxygen species [50]. This normal process is utilized in vitro to denote cell viability and identify the efficacy and downstream mechanisms of novel anticancer drugs [51,52].

One of the aspects that lead to apoptosis is oxidative stress and increased superoxide radicals [53]. The excessive production of ROS causes oxidative stress, mediating damage to the DNA and RNA in addition to protein and lipid oxidization [54]. The results of this study show that treatment with the novel complex **2** can induce apoptosis by increased production of ROS, thus exasperating cytotoxicity. Moreover, to expand on apoptosis ensuing in the cells incubated with this novel complex, mitochondrial membrane potential disruption was usually observed as the early sign of apoptotic death [55]. Consequently, the ΔΨ_m_ assay using JC-1 dye was performed to observe mitochondrial damage, where the reduction in mitochondrial membrane potential was occurring in a concentration-dependent manner. This disruption in ΔΨ_m_ is one of the main contributing factors in cellular apoptosis [55].

There are two signaling pathways that can trigger apoptosis: intrinsic and extrinsic pathways [50]. Caspase 3 is a marker of apoptosis, which is termed executioner caspase [56]. The current study examined the molecular mechanism of action of complex **2** on the induction of the apoptotic pathway in vitro, which was investigated by q-RT PCR analyses. Caspase 3 induction was observed after treatment with complex **2**, which is indicative that breast cancer cells are undergoing apoptosis, as reported in another study that used venetoclax on the same breast cancer cell line, MDA-MB-231 [57].

In the intrinsic apoptotic pathway, the BCL-2 family proteins are regulators in the molecular mechanisms of apoptosis, which is mediated by mitochondria. They are divided based on their function as proapoptotic and antiapoptotic proteins. Any disruption in the homeostasis between these two groups of proteins can affect the fate of cells [58]. The antiapoptotic BCL-2 family has been overexpressed in several human cancers, leading to resistance to apoptosis and chemotherapy [59]. Previous reports showed that the antiapoptotic BCL-2 proteins maintain mitochondrial membrane integrity and prevent cell death by direct interaction with proapoptotic members [60]. BCL-xL is known to be involved in mitochondrial metabolism and suppression of proapoptotic Ca^2+^ signaling, indicating its possible involvement in cancer cell death [61].

In the results obtained with MDA-MB-231 cells, there was a downregulation of BCL-xL gene expression. However, in the 0.3 µM complex **2** treatments, there was an activation of the BCL-xL. Corresponding to this slight elevation, Al-Khayal et al. denoted that the low concentration of novel anticancer agents elevated the expression of BCL-2 protein [62], which occurs initially in response to chemotherapy. Bessou et al. provided evidence that BCL-xL is essential for cell migration by affecting the production of mitochondrial ROS. The inhibition of BCL-xL decreased overall levels of ROS, thus decreasing cell migration, and later, the overexpression of such protein regained the ROS levels and restored cell migration [63].

The gene expression analyses showed that MDA-MB-231 cells treated with complex **2** increased the proapoptotic gene caspase-3 and decreased the expression of the antiapoptotic gene BCL-xL. Thus, we can conclude that this drug can elicit apoptosis in breast cancer MDA-MB-231 cells, attributing it to the upregulation of caspase-3, downregulation of BCL-xL, and alterations in mitochondrial membrane potential. Complex **2** has a higher anticancer potential than cisplatin and was seen to trigger apoptosis and cytotoxicity by eliciting oxidative stress and mitochondrial damage in breast cancer cell lines.

## 5. Conclusions

This report details the synthesis, characterization, and antitumor properties of novel phosphanegold(I) complexes containing 3-Thiosemicarbano-butan-2-one oxime (TBO) as the ligand. Spectroscopic data indicated that the TBO ligands coordinate to the Au(I) center via the sulfur atom of thoine group. Solution studies of complexes **1**–**4** demonstrated the high stability of the complexes in DMSO and H_2_O:EtOH (1:1) solvent system. In vitro cytotoxicity assays revealed that complexes **2**, **3**, and **4** exhibited greater cytotoxicity than cisplatin. Further investigations of complex **2** indicated its potential as an apoptosis-inducing agent in breast cancer MDA-MB-231 cells, with upregulation of caspase-3, downregulation of BCL-xL, and alterations in mitochondrial membrane potential. Taken together, these findings suggest that the newly synthesized complexes hold promise as potent anticancer agents capable of inhibiting cancer cell proliferation. However, further in vitro and in vivo studies are necessary to determine the effect on different cancer cell lines and fully evaluate the efficacy, safety, pharmacokinetics, and potential side effects of the complexes.

## Data Availability

The data are contained within the article or Appendix A.

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
