# Peer review of "Synthesis, Characterization, and Anticancer Activity of Phosphanegold(i) Complexes of 3-Thiosemicarbano-butan-2-one Oxime"

_biomedicines, 2023, doi:10.3390/biomedicines11092512_

Round 1
Reviewer 1 Report
In recent years, cancer has been a scourge of modern society that is global and widespread. All this determines the relevance of the problem under consideration and the significance of the results presented by the authors. This paper presented the synthesis, characterization, and antitumor properties of novel phosphanegold(I) complexes incorporating 3-Thiosemicarbano-butan-2-one oxime (TBO) as the ligand. This is a very interesting, useful and up-to-date article.
The literature cited in this paper is sufficient in volume and content (64). All 5 figures are of sufficiently high quality. The 3 tables are well presented. I recommend that the journal editors accept the article with major corrections.
Dear Author!
Please revisit the article with reg. no.2545771 in accordance with Editor’s comments:
· In line 143-147 “Complexes 1-4 were prepared by combining AgPF6 (0.126g, 0.5 mmol) dissolved in 10 mL ethanol with Me3PAuBr(1) (0.176 g, 0.5 mmol), Et3PAuCl(2) (0.175 g, 0.5 mmol), t-But3PAuCl(3) (0.217 g, 0.5 mmol), Ph3PAuCl(4) (0.247 g, 0.5 mmol) in 10 mL dichloro- methane. The mixture was stirred at room temperature for 15 mins and filtered off to remove AgCl white precipitate.” To obtained the first complex with Me3PAuBr(1), don't you filter out a yellow precipitate of AgBr?
· In line 278-281 Based on the compared IR spectra, you conclude that the NH2 group is not involved in coordination to the metal ion. How will you explain the shift to lower frequencies of this group with 16, 17 or 18 cm-1 in the spectra of the complexes 1-4? The data are given in Table S2 of the supporting information.
· In line 313-314 Table 1 we have 0.75 ppm shifted for one of two protons from NH2 in the complex 1; 0.62 ppm for complex 2; 0.77 ppm – complex 3 and 0.38 ppm – for complex 4. For second protons from NH2 the signal are shifted at: 0.65 ppm; 0.53 ppm; 0.67 ppm and 0.31 ppm, respectively. Don't you think the NH2 group from the free ligand is also involved in the coordination?
· The conclusions in lines 298-301 should be edited. In my opinion, N-atom from the NH2 group is also involved in the coordination with Au(I). These data are in agreement with the data from the IR spectra.
· The conclusions should be edited.
Reviewer 2 Report
Four novel phosphanegold(I) complexes were synthesized and evaluated for their in vitro cytotoxicity against three different cancer cell lines. Complexes 2, 3, and 4 exhibited significant cytotoxic effects against all tested cancer cell lines and showed significantly higher activity than cisplatin. Complex 2 exerts potent anticancer activity against breast cancer cells (MDA-MB-231) through the induction of oxidative stress, mitochondrial dysfunction, and apoptosis. The manuscript is an extension of the previous published manuscript that should be cited in the article (J Med Chem. 2022 Nov 10;65(21):14424-14440. doi: 10.1021/acs.jmedchem.2c00737). The manuscript deserves to be published after the following modifications:
a) The tested compounds should be evaluated on normal cells and selectivity index also reported.
b) Table 3. SD or SE should be reported for each compound. Generally, each concentration was analyzed in triplicate and the experiment was repeated three times.
c) Figure 3. The ROS production was similar increasing the concentration of compound 2, and it was not increase increasing the concentration of compound 2. Can the authors rationalize this effect? The ROS production was almost the same at 0.3 micromolar and and at 10-fold increased concentration.
Round 2
Reviewer 1 Report
I accept the changes made to the article and propose to the editorial board to accept it for publication in the journal Biomedicines.
Reviewer 2 Report
The manuscript can be published in the present form.